# Population Growth Parameters of *Scymnus nubilus* Fed Single-Aphid Diets of *Aphis fabae* or *Myzus persicae*

**DOI:** 10.3390/insects15070486

**Published:** 2024-06-28

**Authors:** Isabel Borges, Guillaume J. Dury, António O. Soares

**Affiliations:** 1cE3c–ABG–Centre for Ecology, Evolution and Environmental Changes and Azorean Biodiversity Group, Faculty of Science and Technology, University of the Azores, 9501-321 Ponta Delgada, Portugal; antonio.oc.soares@uac.pt; 2Department of Biology, University of the Azores, 9501-321 Ponta Delgada, Portugal; gjdury@utexas.edu; 3Department of Integrative Biology, College of Natural Sciences, University of Texas at Austin, 2415 Speedway #C0930, Austin, TX 78712, USA

**Keywords:** aphid, ladybird, predator, biological control, population growth parameters

## Abstract

**Simple Summary:**

Food security and environmental sustainability are two hot topics nowadays. Chemical pesticides used with no discretion threaten human health and ecosystem functioning and, thus, ecological alternatives for herbivorous pest control are required. An environmentally friendly alternative to chemical pesticides is biological control—the use of natural enemies to control pest populations. Ladybirds (known as ladybugs in North America) are iconic biological control agents used against several herbivorous pests, such as aphids and scale insects, among others. However, small-sized ladybirds such as *Scymnus* are still poorly studied, and there is only one case of a biological control program using species in this genus. In the Azores archipelago (Portugal), 30% of ladybird species are small, but present in high population densities; therefore, despite their size, their ability to help to regulate prey population densities is expected to be important. In this work, we studied the life cycle of *Scymnus nubilus* on two new prey species: aphids that attack agricultural and forestry plants. The biological performance of the ladybird predator on the new prey tested was the best obtained to date. Our results agree with other studies focusing on the use of *Scymnus* ladybirds as biocontrol agents and suggest that the predator can contribute to the control of aphid pests.

**Abstract:**

Life tables are an important tool to forecast the performance of biological control agents used in pest management programs, and they are often assessed in terms of population growth. In the present study, the suitability of the aphids *Aphis fabae* Scopoli and *Myzus persicae* (Sulzer) for the ladybird predator *Scymnus nubilus* Mulsant was assessed for the first time. For this, we evaluated and compared the life history traits of immature individuals and adults of the predator fed single-aphid diets and the consequences of the single-aphid diets for the demographic parameters. *Scymnus nubilus* that were fed *A. fabae* were significantly more fecund and presented a shorter immature development time than those fed *M. persicae*. The predators fed *A. fabae* had a significantly higher net reproductive rate, an intrinsic and finite rate of increase, while their doubling time was significantly lower than that of those fed *M. persicae*. The aphid species used in this study are new additions to the essential prey list of the ladybird, with the predator presenting a better biological performance than that found on the previously known essential prey species.

## 1. Introduction

Aphid pest management in orchards relies heavily on chemical control [1,2,3]. Some efforts to promote biological control have been made, but the reduced impacts found on aphid populations have made the task quite challenging [4,5]. As a main pilar of integrated pest management (IPM) programs, biological control can contribute greatly to reduce the quantity of pesticides applied. Concerned with the detrimental effects of such products on human health and the environment, the European Union has funded programs to promote biological control using ladybirds in orchards precisely with the objective of reducing the use of phytopharmaceutical products [6]. Biological control as a component of IPM for crop protection plays an important role for several Sustainable Development Goals of the Food and Agriculture Organization of the United Nations, for instance, to achieve food security, promote sustainable agriculture, and halt biodiversity loss.

Ladybirds (Coleoptera: Coccinellidae) act as natural biological control agents of herbivorous pests; indeed, this is the primary ecosystem service provided by these predator insects [7]. In 1989, the ‘Vedalia Symposium of Biological Control: A Century of Success’ was held in Riverside, California (United States of America), to commemorate 100 years of the outstanding success of the biological control program that saved the citrus industry. Specifically, it commemorated the most successful case of classical biological control, which was the importation of the vedalia ladybird, *Novius cardinalis* (Mulsant), from Australia to control the cottony cushion scale, *Icerya purchasi* Maskell (Hemiptera: Margarodidae), in the citrus orchards of California.

Among Coccinellidae, most research has focused on the biology and ecology of large species. In Mediterranean areas and habitats of small insular regions, however, an important proportion of the ladybird community (30%) is composed of small species weighing less than 5 mg [8,9,10]. Thus, small ladybirds are expected to make an important contribution to the natural control of insect pests. Despite this, small ladybirds, such as most Scymnini, *sensu* [11], have often been overlooked, and their potential for aphid control remains to be assessed (for details, see [12]).

In Portugal, Scymnini are the most abundant ladybirds in citrus [13], olive, and chestnut groves [14]. This group is also common in almond groves [14,15]. In Spain, Scymnini dominate olive groves [16]. Most Scymnini are very small, often measuring 2–3 mm long, with domed pubescent bodies and dull colorations. They also have diverse food preferences, with predators of aphids, scale insects, mealybugs, adelgids, and whiteflies. They may, therefore, serve as biological control agents against a wide range of fruit orchard and forest pests.

The type genus of Scymini, *Scymnus*, is also the largest genus in the tribe, with over eight hundred species [17]. To our knowledge, only one attempt has been made to use *Scymnus* spp. as biological control agents in pest management programs [18]. The attempt aimed to control the hemlock woolly adelgid, *Adelges tsugae* (Annand) (Hemiptera: Adelgidae), but the ladybirds failed to establish after their release [18]. In contrast, aphidophagous Scymnini have yet to be used in a biological control program; however, recent studies on aphidophagous *Scymnus* species indicate that they are efficient predators of aphids present in orchards [19,20,21,22,23]. For instance, *Scymnus subvillosus* (Goeze) and *Scymnus interruptus* (Goeze) show promise in the biological control of the aphids *Aphis spiraecola* Patch (Hemiptera: Aphididae) and *Aphis gossypii* Glover in clementine orchards (*Citrus* × *clementina*; Sapindales: Rutaceae) when they are present early [21]. In the Portuguese archipelago of the Azores, the potential of *Scymnus nubilus* Mulsant to control the aphid pests of endemic plants is being studied, and they have shown promise for the control of *A. spiraecola* on Azorean laurustinus (*Viburnum treleasei* Gand.; Dipsacales: Adroxaceae) [22] and *Aphis frangulae* Kaltenbach on Azores buckthorn (*Frangula azorica* Grubov; Rosales: Rhamnaceae) [23].

However, some aspects of the biology and ecology of aphidophagous *Scymnus* are still poorly understood, including their food relationships and the effects of food suitability on population dynamics. For a biological control agent to be successful, it must quickly achieve high population densities to be able to suppress the pest effectively, which occurs when the predators have favorable environmental conditions, such as high food suitability [24]. A high predation rate and high predator population growth rate are then expected to be drivers of effective pest control [25,26].

In the present study, we assessed the suitability of two aphid pests—the black bean aphid, *Aphis fabae* Scopoli (Hemiptera: Aphididae), and the green peach aphid, *Myzus persicae* (Sulzer) (Hemiptera: Aphididae)—as food sources for the ladybird predator *S. nubilus*. To do this, we reared the ladybirds on single-aphid diets of either species, compared measurements of the life history traits of their immature and adult stages, and evaluated the consequences of their diets on their demographic parameters.

## 2. Materials and Methods

### 2.1. Biological Traits

The ladybirds were collected on São Miguel Island, Azores, Portugal, in a coastal prairie area (Santa Clara: 37°44′17.65″ N, 25°42′1.14″ W) using sweep nets. To obtain eggs, field-collected adults of *S. nubilus* were paired in plastic boxes measuring 5 cm diameter × 2 cm height, with a 2.5 cm diameter mesh-covered hole in the lid for ventilation. All of the experiments were performed at 25° ± 1° C, with 75 ± 5% relative humidity, and a 16L:8D light regime in climatized rooms. Every day, the adults were transferred to new boxes, allowing the eggs laid in the old boxes to be monitored to determine the hatching time. Newly hatched larvae were transferred into their own plastic boxes and provided with a single-aphid diet ad libitum (approximately 20–30 aphids) of either *A. fabae* or *M. persicae*, with a mix of aphid developmental stages. Both aphid species were mass-reared in the laboratory at 15° ± 1° C, 75 ± 5% relative humidity, with a 16L:8D light regime in a climatized room using fava bean (*Vicia faba* L.; Fabales: Fabaceae) as the host plant.

To monitor larval development and survival, the boxes were checked daily and the developmental stage was noted, using exuviae as an indicator of molting, until the emergence of the adults. At emergence, the adults’ fresh weight was recorded, and they were sexed using simple external morphological characteristics, as follows: males have a light-colored head (yellowish), while females have a dark-colored head (dark brown). Pairs (N = 10 per treatment) were formed and kept in separate boxes to study how the two single-aphid diets of *A. fabae* or *M. persicae* affected their reproductive performances. Every day, these lab-reared adults were transferred to new boxes with ad libitum aphids of the same species that they were fed as larvae, and the boxes from the previous day were checked for the presence of eggs to determine the pre-oviposition time. After the females started laying, the number of eggs laid per female was counted daily. Four days after being laid, the eggs were inspected to determine the number of hatched larvae, embryonated eggs (fertile eggs with unsuccessful hatching), and infertile eggs. The females were followed until their death. The males that died were replaced with unpaired lab-reared males. While it would have been ideal to obtain the sex ratios of the progeny to use in life table calculations, rearing over fourteen thousand larvae until they completed development to determine their sex was not practical. Therefore, we used the sex ratios obtained from the parental population instead.

### 2.2. Demographic Parameters

To calculate the intrinsic rate of increase ($r_{m}$), the Euler–Lotka equation was iteratively solved, according to the following equation [27]:$$1=\sum_{x=0}^{\infty }e^{-r_{m}(x+1)}l_{x}m_{x}$$
where x is time (days), $l_{x}$ is age-specific survival, and $m_{x}$ is age-specific female offspring (hatched larvae). The following mathematical formulas were used to calculate the net reproductive rate ($R_{0}$, defined as female offspring per female), mean generation time (*τ*), finite rate of increase (*λ*), and doubling time (*DT*):$$R_{0}=\sum l_{x}m_{x}$$
$$\tau =\frac{\ln R_{0}}{r_{m}}$$
$$\lambda =e^{r_{m}}$$
$$DT=\frac{\ln2}{r_{m}}$$

Considering that population growth parameters are calculated with the data of several females, it is not possible to statistically compare them directly, because only one value is obtained. Jackknife analysis was, therefore, used to obtain the means and standard error (SE) of the demographic parameters. To obtain jackknife estimates, a series of pseudo-values of the variables ($E_{all}$) was obtained by eliminating one female at a time from the dataset. The pseudo-values ($P_{j}$) were calculated as follows:$$P_{j}=n\times E_{all}-(n-1)\times E_{-j};j=1,\ldots ,n$$

The population growth parameters were then compared statistically.

When obtaining jackknife estimates, the omission of a male, of an individual that died as an immature individual, or of an infertile female can result in an estimate of the net reproductive rate (*R*_0_, *j*) of zero for groups of individuals for which the number of (female) offspring per female is clearly non-zero [28,29,30]. Since we used only fertile females, our use of jackknifing is robust to these potential issues. Jackknifing can also result in biologically inconsistent life table values [31], however, the values we obtained were all consistent.

### 2.3. Statistical and Survival Analyses

The egg development time and immature development time of *S. nubilus* were not normally distributed for either single-aphid diet; therefore, the non-parametric Mann–Whitney (MW) test was used to compare these variables. The immature survival of the predators was compared using a chi-square test. Data on the pre-oviposition time of the predators fed *A. fabae* and the male weight of the predators fed *M. persicae* could not be normalized and were compared using the MW test. T-tests were performed to compare female weight, fecundity, fertility, oviposition rate, and oviposition period of *S. nubilus* fed either *A. fabae* or *M. persicae*. Homogeneity of variance was confirmed with Levene’s test for all variables except mean generation time, τ. As we were unable to normalize the distribution of $R_{0}$ for the females fed on *A. fabae*, and due to the heterogeneity of variances of τ, the non-parametric MW test was used to compare these variables. T-tests were used for the other population growth parameters after confirming data normality and homogeneity of variances. The aforementioned statistical analyses were performed with SPSS v. 25 [32], and analysis of sex ratio was performed using exact binomial tests in *R* v. 4.3.2 [33].

Survival analysis was performed using the Online Application for Survival Analysis 2 [34]. For both single-aphid diets, the predator mean lifespan was determined and compared using the Log-Rank test. Kaplan-Meier survival curves and 95% confidence intervals were generated using the package *survival* 3.5-7 [35,36] in *R* v. 4.3.2 [33].

## 3. Results

### 3.1. Biological Traits

No difference in incubation time was found between the diets (MW test: *U* = 291.000; N *_A. fabae_* = 30, N *_M. persicae_* = 22; *p* = 0.336), but the ladybird larvae developed significantly faster on the *A. fabae* diet (MW test: *U* = 247.000; N *_A. fabae_* = 37, N *_M. persicae_* = 35; *p* < 0.001). Both diets allowed most of the immature individuals to complete development with no significant differences in survival rate (Chi-square test: *χ^2^* = 0.3756; *df* = 1; *p* = 0.539955). The adult weight was not influenced by diet both for females (*t*-test: *t* = −1.415; *df* = 42; *p* = 0.164) and males (MW test: *U* = −76.500; N *_A. fabae_* = 11, N *_M. persicae_* = 16; *p* = 0.570). The data indicate that the sex ratio is female-biased on an *A. fabae* diet (32% male, 95% CI [18–50%], *p* = 0.0470), whereas it is even on an *M. persicae* diet (48% male, 95% CI [29–63%], *p* = 0.7359). The females took the same time to reach sexual maturity (MW test: *U* = −59.500; N *_A. fabae_* = 13, N *_M. persicae_* = 12; *p* = 0.288), but a significantly larger number of eggs were laid when the females were fed *A. fabae* (*t*-test: *t* = 2.747; *df* = 18; *p* = 0.013). However, the number of hatched larvae did not differ between the diets (*t*-test: *t* = 1.907; *df* = 18; *p* = 0.073). The data indicate that the oviposition rate (*t*-test: *t* = 1.438; *df* = 18; *p* = 0.168) and oviposition period (*t*-test: *t* = 1.644; *df* = 18; *p* = 0.118) are similar under both food regimes (Table 1).

### 3.2. Demographic Parameters

Concerning population growth parameters, we found that the intrinsic rate of increase (*r_m_*; *t*-test: *t* = 3.524; *df* = 18; *p* = 0.002), the finite rate of increase (*λ*; *t*-test: *t* = 3.519; *df* = 18; *p* = 0.002), and the net reproductive rate ($R_{0}$; MW test: *U* = 15.00; N *_A. fabae_* = 10, N *_M. persicae_* = 10; *p* = 0.008) were significantly higher when the ladybirds were fed *A. fabae*. The doubling time (*DT*) was significantly lower when the adults were fed *A. fabae* (*t*-test: *t* = −3.549; *df* = 9; *p* = 0.002). No significant differences were found in the mean generation time (*τ*) between the two diets (MW test: *U* = 35.00; N *_A. fabae_* = 10, N *_M. persicae_* = 10; *p* = 0.257) (Table 2).

The oviposition pace of the females is similar under the different diets and mirrors the oviposition pattern of other aphidophagous species; moreover, it shows typical triangular fecundity, that is, a rapid increase in the number of eggs laid, with a single oviposition peak, followed by a slow decrease in the rate of oviposition (Figure 1). The reproductive lifespan, that is, the oviposition period, was similar under the *A. fabae* and *M. persicae* diets. Most of the eggs were laid within 90 days, but the females continued laying until their death (Figure 1).

On the *A. fabae* diet, the mean lifespan was significantly longer than on *M. persicae* (Log-Rank test: χ^2^ = 398.77, *p* < 0.001; Figure 2). Across the entire life cycle, the predator showed a higher survival rate when fed *A. fabae*.

## 4. Discussion

Population growth parameters provide important information about biological traits, which is essential for sound decision making in pest management. Most often, a high intrinsic rate of increase (*r_m_*) and net reproductive rate (*R*_0_) are used as indicators of good candidates for biological control agents, because they represent fast predator population growth and high reproductive indices. Following this rationale, our results indicate that *S. nubilus* would be able to control *A. fabae* more efficiently than *M. persicae*.

Moreover, both aphid species meet the criteria to be essential prey species for *S. nubilus*. Essential prey species, under the criteria of [37], are those of high enough quality to support the growth and development of predator larvae and the reproduction by adults. Until the last decade, the essential prey list of *S. nubilus* was limited to *A. gossypii* [38] and *Rhopalosiphum padi* L. (Hemiptera: Aphididae) [19], but recent results indicate that *A. spiraecola* and *Cinara juniperi* (De Geer) (Hemiptera: Aphididae) allow the predator to successfully develop [22] and reproduce (unpublished data [22]), as well as *A. frangulae* [23]. In the Azores archipelago, *S. nubilus* can be found in many types of habitats, such as the following: agroecosystems, greenhouses, coastal prairies, city gardens, and orchards. Because it is present in diverse habitats, it is reasonable to assume that its prey range covers many more species. Indeed, *A. fabae* and *M. persicae* allowed the ladybird predator to successfully develop and reproduce. Therefore, these two species are new additions to its essential prey list.

However, even species on the essential prey list vary in food quality for predators. Food quality is an important determinant of the physiological processes in coccinellids [12], with direct impacts on their developmental rate, survival, and reproductive performance, such as fecundity and fertility (e.g., [39,40,41,42,43,44]). Of all of the aphid species on *S. nubilus*’ prey list, *A. fabae* is the best food resource, followed by *M. persicae*. Conversely, *S. nubilus* performs at its worst when fed *C. juniperi*, an aphid that specializes on cedar (this study, [22,23,38]). A congener of the predator, *S. subvillosus,* also has a good biological performance on *A. fabae* and *M. persicae* as prey, and, similarly, has a lower biological performance when fed *M. persicae* [20]. These differences in performance may represent inherent differences in the quality of the aphid species as food items (e.g., [45]), a tri-trophic interaction between aphids, their host plants, and the predator—where the predators that consume the aphids that are well-suited to their host plant do better themselves [46,47]—or both. Notably, *A. fabae* performs much better on *V. faba* than on *M. persicae* [48]; moreover, both this study and [20] used this host plant to rear both aphid species.

Recent research considers the use of *S. nubilus* under an augmentation approach to control aphids in the Azores (e.g., [22,23]) in light of their life history and demographic traits. *Scymnus* species, particularly *S. nubilus* and *S. interruptus*, are the most abundant aphidophagous ladybird predator in the Azores [10]. *Scymnus nubilus* can singly lay up to 30 eggs per day, scattering the eggs over many aphid colonies—a behavior that may contribute to more effective pest control [19]. The results of [22,23] indicate that *S. nubilus* has the potential to control *S. spiraecola* and *A. frangulae*. The potential of *S. nubilus* to control the aphid species infesting Azorean endemic plants is being further investigated, specifically for the control of the black citrus aphid, *Toxoptera aurantii* (Boyer de Fonscolombe) (Hemiptera: Aphididae). The life table data suggest that *S. interruptus* and *S. subvillosus* perform well on *A. gossypii* and *A. spiraecola*, which are both aphid pests in clementine orchards [21]. We believe that *S. nubilus* could similarly be used to control aphids in citrus orchards.

One downside to using small ladybirds for biological control rather than larger ones is their lower food demands. However, these lower food demands also allow small ladybirds to start feeding earlier on aphid colonies and to feed for longer on senescent prey colonies. Compared to larger species, small ladybirds have an extended oviposition window, because their food requirements are met for longer. Additionally, the higher abundance of small-sized predators could compensate for their lower voracity. After assessing the suitability of aphid species for *Scymnus* ladybirds in the laboratory, field tests are required to assess their effectiveness as biological control agents of aphids in different habitats and field conditions.

## Figures and Tables

**Figure 1 insects-15-00486-f001:**
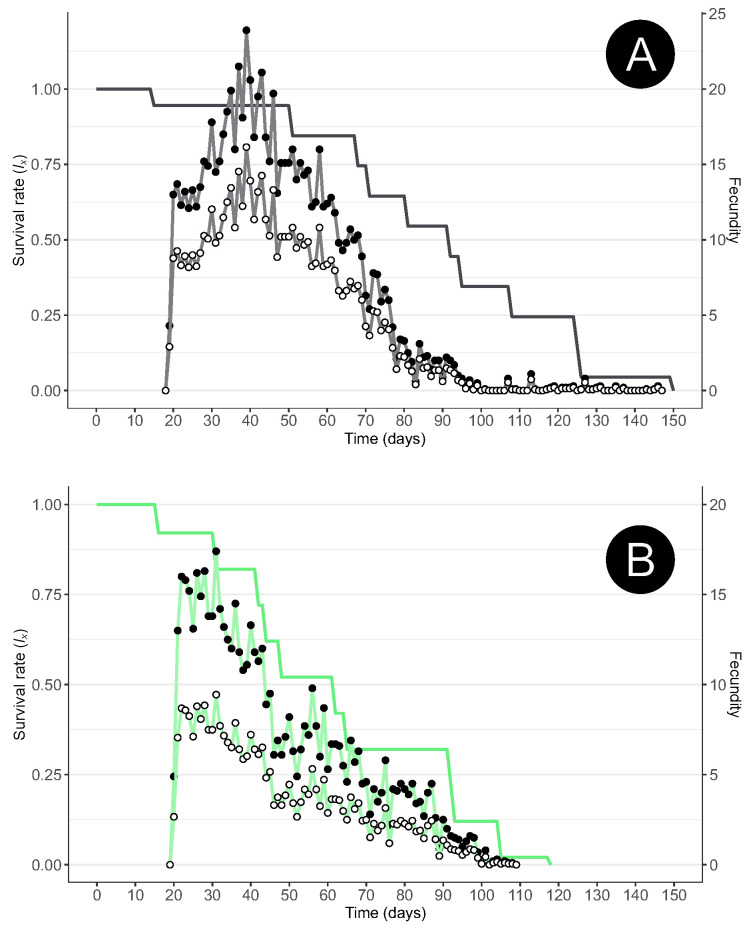
Age-specific survival (*l_x_*; line only), age-specific mean number of offspring (line with full circles), and age-specific mean number of female offspring (*m_x_*; line with white circles) of *Scymnus nubilus* fed (**A**) black bean aphids (*Aphis fabae*) or (**B**) green peach aphids (*Myzus persicae*).

**Figure 2 insects-15-00486-f002:**
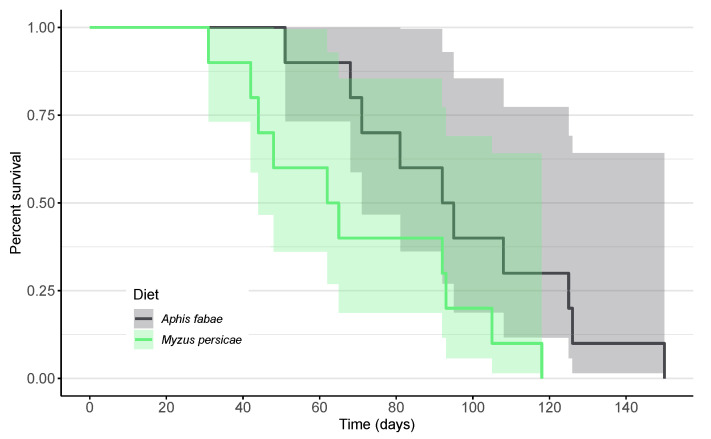
Kaplan–Meier survival curves of *Scymnus nubilus* fed single-aphid diets of black bean aphids (*Aphis fabae*; dark gray) or green peach aphids (*Myzus persicae*; pale green) and 95% confidence intervals.

**Table 1 insects-15-00486-t001:** Biological traits of *Scymnus nubilus* fed either *Aphis fabae* or *Myzus persicae* (25° ± 1 °C, 75 ± 5% relative humidity, and 16L:8D light regime).

	Prey Species	
Predator Biological Trait	*A. fabae*	*M. persicae*
Incubation time (days)	4.2 ± 0.1 a (30)	4.3 ± 0.1 a (22)
Immature development time (days)	11.3 ± 0.1 b (37)	12.1 ± 0.1 a (35)
Immature survival (proportion)	0.97 a (37/38)	0.95 a (35/37)
Pre-oviposition time (days)	4.2 ± 0.2 a (13)	3.6 ± 0.4 a (12)
Fecundity (n. of eggs)	1052.9 ± 98.8 a (10)	654.1 ± 106.4 b (10)
Fertility (n. of fertile eggs)	859.3 ± 88.7 a (10)	607.5 ± 97.8 a (10)
Oviposition rate (n. of eggs/day)	15.4 ± 1.2 a (10)	13.5 ± 0.5 a (10)
Oviposition period (days)	72.9 ± 9.4 a (10)	51.0 ± 9.5 a (10)
Female weight (mg)	1.21 ± 0.02 a (25)	1.24 ± 0.02 a (19)
Male weight (mg)	0.97 ± 0.02 a (11)	0.97 ± 0.02 a (16)
Sex ratio (% F:M)	67.5 * (25:12)	54.3 (19:16)

Where applicable, means are followed by standard error (±SE) and (N). Means and proportion followed by different letters are significantly different. Sex ratio was compared to the null expectation of males and female proportions being equal; asterisk (*) indicates significant (*p* < 0.05) difference from the null.

**Table 2 insects-15-00486-t002:** Population growth parameters (mean ± SE) (*r_m_*: intrinsic rate of increase; λ: finite rate of increase; *DT*: doubling time; *τ*: mean generation time; *R*_0_: net reproductive rate) of *Scymnus nubilus* fed single-aphid diets of *Aphis fabae* or *Myzus persicae* (25° ± 1° C, 75 ± 5% relative humidity, and 16L:8D light regime).

Population Growth Parameters	*A. fabae*	*M. persicae*
*r_m_* (female/female day^−1^)	0.190 ± 0.002 a	0.178 ± 0.003 b
*λ* (female/female day^−1^)	1.210 ± 0.003 a	1.195 ± 0.003 b
*DT* (days)	3.642 ± 0.042 b	3.893 ± 0.057 a
*τ* (days)	32.6 ± 0.7 a	30.0 ± 1.3 a
*R*_0_ (female offspring/female)	489.6 ± 63.6 a	201.2 ± 51.2 b

Population growth parameters followed by different letters are significantly different.

## Data Availability

The original data presented in the study are openly available in Mendeley Data at DOI 10.17632/c5stf3cnyp.1.

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
