# Peer review of "Population Growth Parameters of *Scymnus nubilus* Fed Single-Aphid Diets of *Aphis fabae* or *Myzus persicae"

_insects, 2024, doi:10.3390/insects15070486_

Round 1

Reviewer 1 Report

Comments and Suggestions for Authors

The present paper presents the results of a small-scale laboratory experiment studying the life tables of an aphidophagous ladybird of the genus Scymnus, native to the Azores. Overall, the study appears to be well done, albeit replication is low, and the data are appropriately analyzed (but see comments below). The paper is well written and figures and tables are appropriate.

However, I do have some concerns regarding the description of the methodology. More detail is needed here for a full understanding of the experiments performed. I will list a number of issues in my detailed comments below. Also, I feel that the title does not really cover the content of the study. In fact, this is a small life table study and the data do not allow to say anything much about the control potential of Scymnus nubilus against aphid pests in orchards. So, a less ambitious sounding title would be more fitting in my view.

Detailed comments:

-          Line 65-66: does this statement refer to classical biological control or augmentation? (or both?)

-          72: I don’t think you can state that research on ladybirds has overlooked Scymnini – perhaps yes, if you only refer to aphidophagous species, but there is a considerable body of literature on coccidophagous species like Cryptolaemus montrouzieri, and others. A slight rewrite of the sentence would solve the problem.

-          101-102: “population system…”, this line needs to be revised

-          111: field collected where?

-          113: provide an indication of variation for this set temperature of 25°C; were experiments done in a climatic cabinet?

-          116: how many larvae were tested for the developmental experiment? I have not seen any information on replication for this experiment in the paper

-          117: what stages of the aphid prey were provided? Young nymphs, late instar nymphs, adults, a mixture?

-          128: did you check the sex of the progeny? It is in fact this sex ratio that should be used in life table calculations, rather than the sex ratio of the parental population

-          174: I assume that this line refers to the embryonic time of the eggs laid by the field collected adults – so this cannot be explained by any effect of the tested prey as eggs don’t eat. Did you check whether the diet of the 20 monitored adult couples receiving either aphid prey species affected embryonic time of their eggs? This would be more relevant.

-          Table 1: immature survival is stated to be expressed in %, but the data in the table are clearly proportions

-          Figure 1: “(A)” and “(B)” are each mentioned twice in the caption

-          232-233: is this statement (A. fabae being a better prey than M. persicae) solely based on the results of the present study?

-          239, “better nutrition”: does this imply that prey quality is primarily a matter of providing sufficient nutrients, or do antinutritional components also play a role? (I assume that they do!)

-          The final paragraphs discuss about the potential role of S. nubilus for biological control agents against aphid pests but they do not explain how this ladybird will be “used”. Does this all refer to an augmentative approach in which the ladybird would be mass reared and released? Or do the authors see alternative approaches to its functioning in biological control?

Reviewer 2 Report

Comments and Suggestions for Authors

Borges et al. evaluted population growth rate of Scymnus nubilus feeding on two aphid species: A. fabae and M. persicae, only at 25°C. Overall, the paper is well written and organized, however, it still needs important improvements to be considered.

Introduction: talk more about the importance of population growth in biological control

My main concern is materials and methods where some parts are missing (see the attached pdf). The experimentral protocle is not clear. Did you use individual experiments? I mean one male+one female of S. nubilus with aphids. How many numbers of aphid individuals of each species did you use?

Statistical analysis is quite good but you need to add few things.

Discussion: the authors did not discuss the parameters of population growth. IT is important to mention what is each parameter (R0, TC, …..) for?

two papers about the topic of population growth

Yu JZ, Chi H, Chen BH (2013) Comparison of the life tables and predation rates of Harmonia dimidiata (F.) (Coleoptera: Coccinellidae) fed on Aphis gossypii Glover (Hemiptera: Aphididae) at different temperatures. Biol Control 64:1-9

Ismail M, van Baaren J, Briand V, Pierre JS, Vernon P, Hance T (2014) Fitness consequences of low temperature storage of Aphidius ervi. Biocontrol, 59: 139-178. Doi 10.1007/s10526-013-9551-x.

More specific comments in the pdf.

Comments on the Quality of English Language

Author Response

The authors would like to thank the reviewer for the valuable comments and suggestions. Changes were made to the manuscript accordingly. Answers are provided after each comment. Please, find answers to specific comment in the pdf file.

Introduction: talk more about the importance of population growth in biological control

Text was added. Please check lines 94-96.

My main concern is materials and methods where some parts are missing (see the attached pdf). The experimentral protocle is not clear. Did you use individual experiments? I mean one male+one female of S. nubilus with aphids. Yes. To clarify, text was added. Please check line 122. How many numbers of aphid individuals of each species did you use? Food was provided “ad libitum” as mentioned in the text (line 113).

Statistical analysis is quite good but you need to add few things.

Discussion: the authors did not discuss the parameters of population growth. IT is important to mention what is each parameter (R0, TC, …..) for?

Text was added. Please check lines 216-221.

two papers about the topic of population growth

Yu JZ, Chi H, Chen BH (2013) Comparison of the life tables and predation rates of Harmonia dimidiata (F.) (Coleoptera: Coccinellidae) fed on Aphis gossypii Glover (Hemiptera: Aphididae) at different temperatures. Biol Control 64:1-9

Ismail M, van Baaren J, Briand V, Pierre JS, Vernon P, Hance T (2014) Fitness consequences of low temperature storage of Aphidius ervi. Biocontrol, 59: 139-178. Doi 10.1007/s10526-013-9551-x.

 The references were included in the manuscript.

Round 2

Reviewer 2 Report

Comments and Suggestions for Authors

I read the paper for the second time. the authors have improved their MS. However, they need to consider some more comments attached in the pdf.

Comments on the Quality of English Language

Author Response

Thank you very much for taking the time to read the revised manuscript. Substantial changes were made to the text to “adequately describe methods” and “clearly present the results”. Following comments of the reviewer 2, new text was added to include information that was missing, as pointed out. We appreciate the comments very much because we believe they have contributed for the improvement of the manuscript. Given that Moderate editing of English language was required, a native English speaker colleague has revised the manuscript. We now hope you’ll find the manuscript to your satisfaction. However, if the editor(s) or reviewer still considers necessary to improve the English, we are willing to use English Editing Services. 

Please find the responses to the reviewer comments in the pdf file and the corresponding revisions and corrections in track changes in the Word file in the re-submitted files.
